# Gaitor: Learning a Unified Representation Across Gaits for Real-World Quadruped Locomotion

**Alexander L. Mitchell**[*1,2], **Wolfgang Merkt**[2], **Aristotelis Papatheodorou**[2],
**Ioannis Havoutis**[2], **Ingmar Posner**[1]

[1]Applied AI Lab, [2]Dynamic Robot Systems Control
University of Oxford

**Abstract:** The current state-of-the-art in quadruped locomotion is able to produce a variety of complex motions. These methods either rely on switching between a discrete set of skills or learn a distribution across gaits using complex black-box models. Alternatively, we present *Gaitor*, which learns a disentangled and 2D representation across locomotion gaits. This learnt representation forms a planning space for closed-loop control delivering continuous gait transitions and perceptive terrain traversal. *Gaitor's* latent space is readily interpretable and we discover that during gait transitions, novel unseen gaits emerge. The latent space is disentangled with respect to footswing heights and lengths. This means that these gait characteristics can be varied independently in the 2D latent representation. Together with a simple terrain encoding and a learnt planner operating in the latent space, *Gaitor* can take motion commands including desired gait type and swing characteristics all while reacting to uneven terrain. We evaluate *Gaitor* in both simulation and the real world on the ANYmal C platform. To the best of our knowledge, this is the first work learning a unified and interpretable latent space for multiple gaits, resulting in continuous blending between different locomotion modes on a real quadruped robot. An overview of the methods and results in this paper is found at https://youtu.be/eVFQbRyilCA.

**Keywords:** Representation Learning, Learning for Control, Quadruped Control

## 1 Introduction

Recent advances in optimal control [1, 2, 3, 4, 5] and reinforcement learning (RL) [6, 7, 8, 9, 10] have enabled robust quadruped locomotion over uneven terrain. This has made quadrupeds a promising choice for the execution of inspection, monitoring and search & rescue tasks. Current state-of-the-art methods such as [11, 7] are highly capable and produce a variety of complex motions. However, these methods tend to be either engineered *top-down*, incorporating expert knowledge of the locomotion process via appropriate inductive biases, or *bottom-up* in a data-driven way. Top-down approaches (e.g. [6, 12, 13]) treat gaits as independent skills and are unable to leverage correlations between the skills via an efficient learnt representation and cannot interpolate between the prescribed skill set. In contrast, bottom-up methods tend to be black-box models capturing a blend of locomotion skills as a multi-modal distribution over gaits learnt using a large transformer model [11]. Such transformer-based methods tend to run either at low planning frequencies (less than $100\,\mathrm{Hz}$) or require additional compute for inference. While successful in terms of delivering locomotion capability, such models are typically uninterpretable.

To design a learning-based and interpretable method capable of interpolating between gaits, we take inspiration from *VAE-Loco* [14, 15]. *VAE-Loco* demonstrates that gait-phase relationships emerge automatically when learning a disentangled latent representation based on a limited set of demonstrations. In this paper we present *Gaitor*, a data-driven approach to learning and exploiting a single, unified representation of locomotion dynamics in a disentangled latent space across multiple gaits.

---

[*]Email: mitch@robots.ox.ac.uk

8th Conference on Robot Learning (CoRL 2024), Munich, Germany.

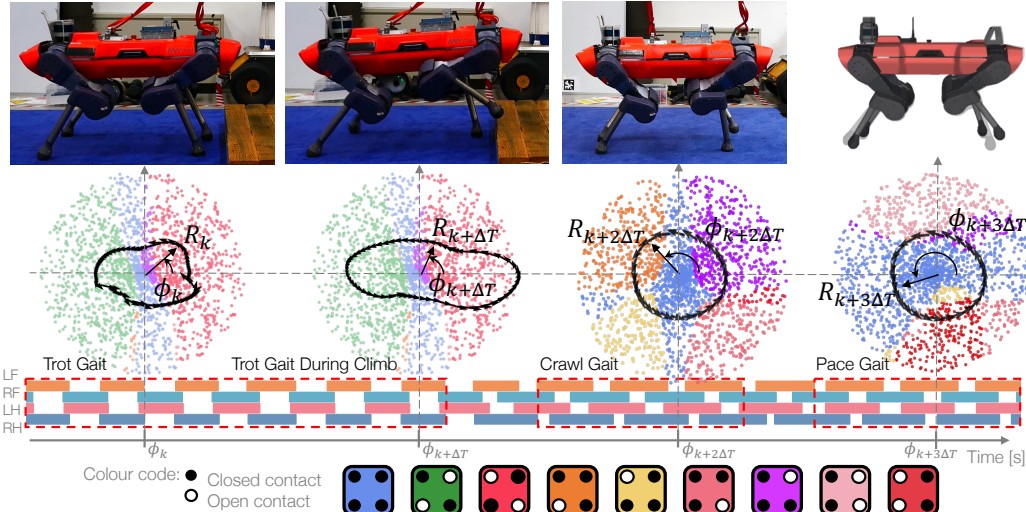

Figure 1: *Gaitor's* structured latent space forms a planning space in which trajectories for trotting on flat ground, climbing terrain with perception, transitioning to crawl and to pace can be achieved on the real robot. The structure of the latent space during these four scenarios are depicted in the second row. The coloured points in the latent space represent the contact state of the robot and a colour code is provided in the bottom row. The black trajectories in the latent space are parameterised by the gait phase $\phi_k$ and radius $R_k$ at timestep $k$. The latent space trajectory adjusts as the robot climbs terrain to change the swing characteristics. As the robot changes gait, the structure of the latent space changes as does the contact schedule via automatically discovered intermediary gaits.

While our work is located in the domain of quadruped locomotion, in contrast to other work in this area our overarching goal is not in the first instance to achieve state-of-the-art performance, but to explore the challenges and opportunities afforded by learning a representational space to encode the synergies between separated locomotion skills. Our primary contribution is to show that a disentangled and ultimately 2D representation can effectively capture quadruped dynamics resulting in continuous gait transition and robust terrain traversal. We discover that gait transitions occur via inferred intermediary gaits not included in the training dataset. Inspection of the learnt representation reveals that *Gaitor* infers the relative gait phase between each skill and this is responsible for the transitionary contact schedules between the gait types. *Gaitor* incorporates a terrain encoding, which adapts the latent space structure in response to the topography beneath the robot in a readily interpretable way. In particular, the swing characteristics are encoded differently in response to changes in terrain. *Gaitor* couples these representations with a learnt planner operating in the latent space, and results in robust terrain-aware trajectories capable of traversing significant obstacles.

To the best of our knowledge, *Gaitor* is the first work that demonstrates how a data-driven approach can create a unified representation for multiple locomotion modes via an interpretable and *disentangled* latent space. The contact states associated with distinct gaits are evident in the latent space. As an operator demands a change in gait, the latent space restructures as shown in Fig. 1. Gait transitions arise since the learnt representation captures meaningful correlations across the gait types resulting in a 2D planning manifold. In our experiments, *Gaitor* is deployed as a realtime controller operating at $400\,\mathrm{Hz}$ on the physical ANYmal C robot. We show continuous gait transitions *on-demand* from trot to crawl to a crawl/pace hybrid and vision-aware climbing onto a $12.5\,\mathrm{cm}$ platform.

## 2 Related Work

Reinforcement learning (RL) [8, 16, 10] is an increasingly popular choice for learning locomotion policies due to advances in simulation capabilities. A recent trend for RL is to solve locomotion problems by learning a set of discrete skills with a high-level planner arbitrating which skill is deployed at any one time. For example, [6] learns a set of 5 distinct skills and a high-level planner to

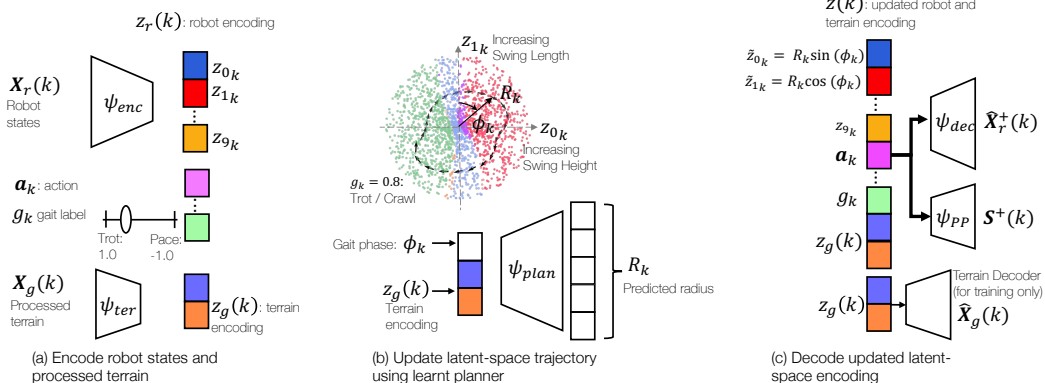

(a) Encode robot states and processed terrain

(b) Update latent-space trajectory using learnt planner

(c) Decode updated latent-space encoding

Figure 2: There are three components to *Gaitor*. The first is a variational autoencoder (VAE) to encode the robot states to build the latent space and is comprised of an encoder $\psi_{enc}$ and decoder $\psi_{dec}$, see panels (a) and (c). The second is a terrain encoder $\psi_{ter}$ which represents a rudimentary encoding of the ground ahead of the robot. A learnt planner $\psi_{plan}$ creates trajectories in the latent space, see panel (b). The planner's trajectory exploits the latent-space structure to vary the robot's gait characteristics (footswing heights and lengths) in response to terrain. The updated latent-space trajectory is decoded to predict a set of future robot states $\mathbf{X}_r^+(k)$ and contact states $\mathbf{S}^+(k)$ using the performance predictor $\psi_{PP}$, see panel (c). The future trajectory is sent to a whole-body controller.

select the sub-skill to traverse the terrain ahead of the robot. These sub-skills include jumping across a gap or climbing over an obstacle. This method produces impressive behaviour, but no information is shared across tasks, meaning that each skill learns from scratch and correlations across skills are unused. Another impressive work *Locomotion-Transformer* [11] learns a generalist policy for multiple locomotion skills for traversing an obstacle course. This approach uses a transformer model to learn a multi-modal policy conditioned on terrain. In contrast, *Gaitor* learns a disentangled 2D representation for the dynamics of multiple locomotion gaits. The space infers the phase relationships between each leg and embeds these into the latent space structure. Unseen intermediate gaits are automatically discovered by traversing this structured latent space.

Work by [13] tackles gait switching by learning to predict the phases between each leg. The predicted phases create a contact schedule for a model predictive controller (MPC), which computes the locomotion trajectory. In [13], the gait phase is an inductive bias designed to explicitly determine the swing and stance properties of the gait. Similarly to *Gaitor*, work by [12] conducted contemporaneously to *Gaitor*, uses a latent space in which different gaits or skills are embedded. A gait generator operates from this space and selects a particular gait to solve part of the locomotion problem. This uses a series of low-level skills to learn the latent representation using pre-determined contact schedules. In contrast, *Gaitor* infers the relationships between gait types and automatically discovers the intermediary gaits needed for smooth transitions from discrete examples.

Locomotion planning in learnt latent representations is a growing field of interest. For example, [17] and [18] use architectural biases to enforce Fourier-style dynamics in latent space for locomotion. The former method is designed to create a compact representation for character animation and the latter for control of a biped robot in simulation. *VAE-Loco* [14] which forms inspiration *Gaitor* infers rotational dynamics to represent locomotion in a structured latent space. However, the rotational dynamics in *VAE-Loco's* latent space arise from representing the contact dynamics in the space rather than using Fourier-inspired architectures. *VAE-Loco* is constrained to a single gait and blind operation on flat ground only. Alternatively, *Gaitor* learns a single, unified latent representation for quadruped locomotion across *multiple* gaits to achieve perceptive locomotion in uneven terrains.

## 3 Gaitor

This section outlines the components of *Gaitor*. A variational autoencoder (VAE) [19, 20] similar to *VAE-Loco* [14] and augmented with a learnt terrain encoder is exposed to distinct gaits, trot, pace

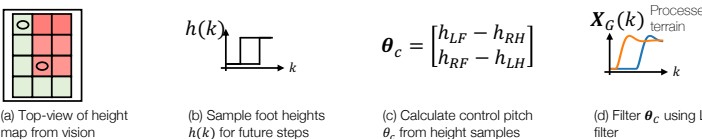

(a) Top-view of height map from vision

(b) Sample foot heights $h(k)$ for future steps

(c) Calculate control pitch $\theta_c$ from height samples

(d) Filter $\boldsymbol{\theta}_c$ using LTI filter

Figure 3: Terrain processing pipeline. The onboard perception module produces a 2.5D height map of the terrain around the robot. This map is sampled at the footfall positions to estimate the heights of the terrain at these locations, see panels (a) and (b). The control-pitch angle $\theta_c$ is defined as the difference in heights between the front and rear footfalls, see panel (c). These values are input to a second-order filter to create the terrain's encoder input $\mathbf{X}_G$ (panel (d)).

and crawl. This creates a shared representation across all the gaits and is conditioned on a learnt terrain encoding. Once the representation is created, a learnt planner is trained and operates in the latent space. This planner adjusts the robot's gait characteristics in response to changing terrain. The complete architecture is illustrated in Fig. 2.

## 3.1 Learning a Unified Representation

Training the VAE and terrain encoder creates the latent representation. The robot state at time step $k$ is $\mathbf{x}_k = [\mathbf{q}_k, \mathbf{ee}_k, \tau_k, \lambda_k, \dot{\mathbf{c}}_k, c_{\theta_x}, c_{\theta_y}, \Delta\mathbf{c}_k]$ and represents the joint angles, end-effector positions in the base frame, joint torques, contact forces, base velocity, roll and pitch of the base and the base-pose evolution respectively. The VAE's encoder $\psi_{enc}$ takes a history of these $N$ states as input sampled at a frequency of $f_{enc}$ creating $\mathbf{X}_r(k)$, see Fig. 2 (a). The decoder output $\hat{\mathbf{X}}_r^+(k)$ is a trajectory of $M$ future robot states from time step $k$ sampled using a frequency $f_{dec}$. The encoder tracks the evolution of the base pose $\Delta\mathbf{c}_k$ from a fixed coordinate. This coordinate is the earliest time point in the encoder input and all input poses are relative to this frame. All pose orientations are represented in tangent space. This is a common representation for the Lie-Algebra of 3D rotations since composing successive transformations using a Euclidean or a Quaternion representation is non-linear, but in tangent space, this composition is linear, see [21, 22] for examples. The output of the encoder parameterises a Gaussian distribution to create the robot latent sample $\mathbf{z}_r(k)$.

The decoder predicts the future robot states $\hat{\mathbf{X}}_r^+(k)$ using the inferred latent variable and user-controlled inputs $\mathbf{a}_k$ and $g_k$ as illustrated in Fig. 2 (c). The operator can control the robot's base velocity heading using the action input $\mathbf{a}_k$ and the gait using the gait label $g_k$. The predicted base-pose evolution is expressed relative to the base pose at the previous time-step. The inputs to the decoder are $\mathbf{z}_r(k)$, the desired base-heading action $\mathbf{a}_k$, the desired gait label $g_k$, and terrain encoding $\mathbf{z}_g(k)$. Importantly, due to our formulation of learning a unified latent space across gaits, $g_k \in \mathbb{R}$ is a *continuous* value used to select a gait or intermediate gait type. Distinct gaits such as trot, crawl, and pace are selected when $g_k = \{1, 0, -1\}$ respectively. The performance predictor $\psi_{PP}$ estimates the contact states of the feet $\mathbf{S}(k)$ and takes the robot encoding as well as the desired action and gait as input, see Fig. 2 (c). The contact states are predicted during operation and sent to the tracking controller to enforce the contact dynamics.

## 3.2 Terrain Encoder

The terrain encoding is required so that the decoder and planner can react to terrain height variations. The robot's perception module provides a 2.5D height map of the terrain. This is constructed using four depth cameras on the robot. Complete details for creating the depth map from a depth field are found in [23, 24]. This raw map is filtered and in-painted in a similar fashion to [25]. The height map is sampled at the locations of the next predicted footholds and the heights $\mathbf{h}(k)$ are processed to create the terrain encoder's input $\mathbf{X}_g(k)$. The processing pipeline takes the heights of the future footholds and finds the differences between left front, right hind and right front and left hind respectively. We define these relative differences as the control pitch $\theta_c$, see Fig. 3 (c). The control pitch varies discontinuously and updates when the robot takes a step. To convert this discontinuous variable to a continuous one, we filter it using a second-order linear time-invariant

(LTI) filter. This converts the step-like control pitch sequence into a continuous time varying signal $\mathbf{X}_g(k)$ of the type shown in Fig. 3 (d). This is achieved by setting the LTI's rise time to occur when the foot in swing is at its highest point and using a damping factor of $0.5$. The input to the terrain encoder $\psi_{\text{ter}}$ is a matrix of $N$ stacked values $\mathbf{X}_g(k)$ sampled at the control frequency $f_c$. The output of the terrain encoder $\mathbf{z}_g(k)$ forms part of the decoder's input $\tilde{\mathbf{z}}(k)$ as shown in Fig. 2 (c). During training $\mathbf{z}_g(k)$ is input to a terrain decoder used to reconstruct a prediction of $\mathbf{X}_g(k)$. This element is discarded after training.

### 3.3 Training the VAE and Terrain Encoder

The VAE and performance predictor are optimised using the evidence lower bound (ELBO) and a binary cross entropy (BCE) loss. The performance predictor estimates which feet are in contact $\mathbf{S}^+(k)$ given the robot-state input $\mathbf{X}_r(k)$. The BCE loss optimises the performance predictor's parameters and gradients from this loss pass through the VAE's encoder similarly to *VAE-Loco* [14]. The ELBO loss used here takes the form of

$$\mathcal{L}_{\text{ELBO}} = \text{MSE}(\mathbf{X}_r^+(k), \hat{\mathbf{X}}_r^+(k)) + \beta D_{\text{KL}}[q(\mathbf{z}|\mathbf{X}_r(k))||p(\mathbf{z})], \tag{1}$$

and is summed with the BCE loss creating

$$\mathcal{L}_{\text{VAE}} = \mathcal{L}_{\text{ELBO}} + \gamma \text{BCE}(\mathbf{S}^+(k), \hat{\mathbf{S}}^+(k)). \tag{2}$$

The VAE is optimised using GECO [26] to tune $\beta$ during training. The terrain encoder's weights are updated using the MSE loss between predicted $\hat{\mathbf{X}}_g(k)$ and the ground-truth data $\mathbf{X}_g(k)$ and gradients from the VAE decoder's MSE loss. The terrain autoencoder loss is therefore

$$\mathcal{L}_{\text{TER}} = \text{MSE}(\mathbf{X}_r^+(k), \hat{\mathbf{X}}_r^+(k)) + \text{MSE}(\mathbf{X}_g(k), \hat{\mathbf{X}}_g(k)). \tag{3}$$

The terrain decoder is only used for training purposes and is not required for deployment.

### 3.4 Latent Space Planner

The planner produces latent space trajectories as depicted in Fig. 2 (b) and is designed to adjust the robot's swing lengths and heights to help the robot traverse terrain. It is discovered in Sec. 4.1 that the footswing heights and lengths are encoded into two dimensions of latent space $z_0$ and $z_1$. Therefore, we design the planner to exploit the disentangled space by only adjusting $z_0$ and $z_1$. The planner is parameterised in polar coordinates and takes the gait phase $\phi(k)$ and terrain latent $\mathbf{z}_g(k)$ as input to predict the radius of the elliptical trajectory $R(k)$, see Fig. 2 (b). This trajectory updates the values of the two latent dimensions so that

$$\tilde{z}_0(k) = R(k)sin(\phi(k)), \qquad \text{and} \qquad \tilde{z}_1(k) = R(k)sin(\phi(k) + \pi/2). \tag{4}$$

To ensure the planner reproduces these variations to a high fidelity, an architecture inspired by [27] is used as seen in Fig. 2 (b). The planner predicts a probability distribution $r_c$ for the radius over $C$ discrete values via a softmax. A weighted sum of the predicted probabilities multiplied by the bin values estimates the radius $R(k)$.

### 3.5 Training the Planner

The planner can be trained using any framework such as an online or offline RL method, or a supervised learning approach. Here, behavioural cloning is used to train the planner for simplicity. Expert trajectories are encoded into the trained VAE producing latent trajectories, $\mathbf{z}_0^*, ..., \mathbf{z}_D^*$, where $D$ is the number of time steps in the encoded episode. Only the dimensions $z_0$ and $z_1$ are retained to create the dataset for the planner and are converted to polar coordinates. Thus, $z_0^*(0), ..., z_0^*(D)$ and $z_1^*(0), ..., z_1^*(D)$ are converted to radii $R^*(0), ..., R^*(D)$, and phases $\phi(0), ..., \phi(D)$. The relationships between gait phase, radius and latent trajectories are

$$\phi(k) = \text{atan2}(z_0^*(k), z_1^*(k)) \tag{5}$$

$$R^*(k) = \sqrt{z_0^{*^2}(k) + z_1^{*^2}(k)} \tag{6}$$

The planner predicts the radius $R(k)$ given the gait phase $\phi(k)$ and the terrain encoding $\mathbf{z}_g(k)$ as inputs. The total loss is the sum between the MSE to reconstruct the target radius $R^*(k)$ and the cross-entropy loss between the prediction $r_c$ and the label $r^*$:

$$\mathcal{L}_{\text{PLAN}} = \text{MSE}(R(k), R^*(k)) - \sum_{c=0}^{C-1} \log \left( \frac{\exp(r_c)}{\sum_{i=0}^{I-1} \exp(r_c)} r^* \right). \tag{7}$$

### 3.6 Deployment of Gaitor

During deployment, the robot and terrain encodings are estimated using the VAE and terrain encoders respectively. The operator chooses the base velocity action $\mathbf{a}_k$, the desired gait $g_k$ and the gait speed $\Delta\phi_k$. The gait speed $\Delta\phi_k$ controls the robot's cadence (number of footfalls per minute) and updates the gait phase every time step as

$$\phi(k) = \phi(k-1) + \Delta\phi_k. \tag{8}$$

The latent space planner uses the updated phase and the terrain encoding to predict the latent radius. This is used to overwrite dimensions $z_0$ and $z_1$ in the robot encoding $\mathbf{z}_r(k)$ as per Eq. (4). The updated encoding $\tilde{\mathbf{z}}(k)$ is formed by concatenating the updated robot encoding, the terrain encoding, action and gait label. The predicted joint angles, torques and future base-pose are extracted from $\hat{\mathbf{X}}^+(k)$ and are sent to a whole-body controller (WBC) [28]. During our deployment, we discovered a phase-lead is required for best base-orientation tracking. Therefore, we set the robot's base pitch to the average of the control pitch $\theta_c$.

## 4 Experimental Results

Our evaluation of *Gaitor* is designed to answer the following guiding questions: 1) What does the structure of the latent space look like for multiple gaits and how do transitions occur? 2) How does the latent space structure and *Gaitor* adjust the robot's gait characteristics to changes in terrain? 3) How do *Gaitor's* trajectories compare to the expert's ones? Please refer to Sec. A.4 for *Gaitor's* hyperparameters and Sec. A.2 for dataset generation. Experimental results are found in this video.

### 4.1 Latent Space Structure Results in Gait Switching

*Gaitor's* latent space structure is investigated to understand how multiple gaits are encoded. The model used in all the experiments has a latent dimension of 10 units and the space is investigated to reveal which dimensions encode quantities useful for locomotion planning. To do this, we inject oscillatory trajectories into each dimension of the latent space and decode them. Visualising the decoded results reveals that just two oscillations in latent space dimensions ($z_0$ and $z_1$) can reconstruct complete locomotion trajectories. These latent dimensions have the lowest variance and control the robot's footswing lengths

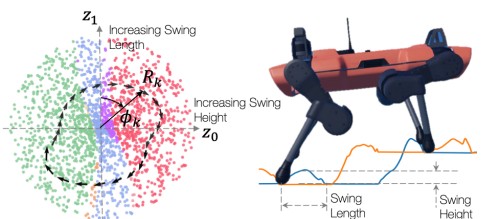

Figure 4: The latent space is disentangled and structured such that displacements in the horizontal $z_0$ axis increase the robot's step height, whereas movements in the vertical axis $z_1$ correlate to the robot's swing length.

and heights independently. The swing length is in the direction of the desired base heading. Fig. 4 shows the 2D slice made by plotting $z_0$ along the horizontal and $z_1$ along the vertical. The points in this plane are colour coded by contact state, where green corresponds to left front and right hind. Red is the opposite contact state (right front and left hind), and blue represents all feet in contact.

We operate *Gaitor* as a planner in a closed-loop control framework on the real robot and command different gait types. The trajectories injected into latent space are two dimensional as described in Eq. (4) and are only injected into dimensions $z_0$ and $z_1$. This results in the robot changing gait continuously from trot to crawl and then to crawl/pace hybrid. The crawl/pace gait is characterised by a contact schedule where the hind foot makes contact as the front foot breaks contact simultaneously.

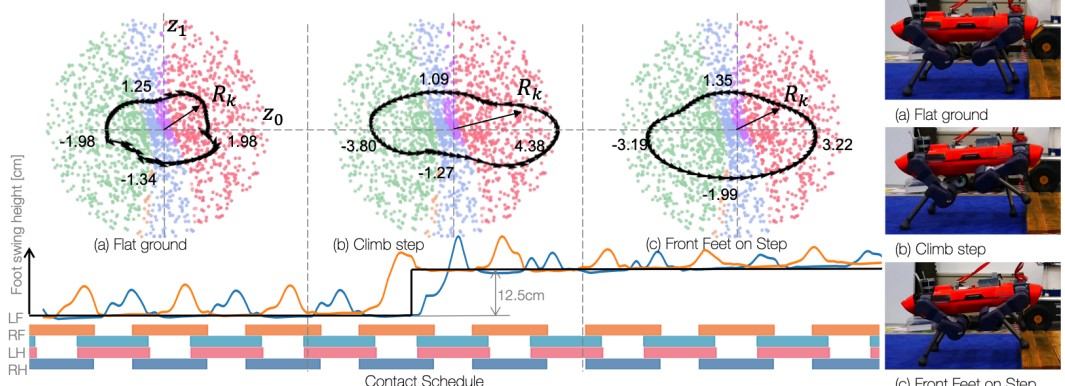

Figure 5: *Gaitor* deployed on the robot climbing a $12.5\,\mathrm{cm}$ platform. The planner and latent-space structure act in concert to alter the locomotion trajectories in response to the terrain. The first phase is flat ground where the planner produces a circular ellipse. When the robot steps onto the platform, the planner radius increases dramatically along the horizontal axis, but is reduced in the vertical axis compared to flat ground operation. This results in shorter footstep heights, but longer swing distances. Once both front feet are on the terrain, the latent-space trajectory is circular. The latent-space trajectory synchronised with the robot climbing terrain can be found in this video.

Note that the expert pace gait is unstable on the real robot so is not deployed in this paper. Only a 2D oscillation is required for *Gaitor* to generate gait switching.

To visualise how the gaits are encoded in latent space, the two latent dimensions $z_0$ and $z_1$ are plotted in Fig. 1 as the operator commands different gait types. Each point in Fig. 1 is colour coded to indicate which feet are in contact. As the gait label $g_k$ is varied from $1.0$ (trot) to $-1.0$ (pace), the latent space structure changes in concert with the contact schedule shown in the row below. It is observed that *Gaitor* produces gait switching in a particular order: trot to crawl to pace and the reverse only. To investigate why gait transitions occur in this order, we observe the phase relationships between legs during transition. During trot, the left front (LF) and left hind (LH) are out-of-phase with one another in the contact schedule (see row 3 of Fig. 1): the LF is in contact as the LH is in swing. However during pace, the LF and LH are in phase as these feet are swinging together. As the operator requests a gait transition from trot to pace or pace to trot, the crawl gait representation and contact schedule emerges at the midpoint of the transition, while the base velocity action is used to reduce the robot's velocity before transitioning into pace safely. This is because the phase between the LF and LH is midway between in-phase and out-of-phase for the crawl gait: only one foot swings at once during crawl. *Gaitor* is able to infer this relationship which is encoded as an inductive bias to represent multiple different gaits in the representation efficiently. *Gaitor* finds intermediary latent space structures to represent the contact schedules of the transitionary gaits e.g. trot/crawl or crawl/pace, see panel (b) and (d) in Fig. 7 and these produce smooth and continuous gait transitions. The ordering of the gaits in latent space using their in-phase relationships is similar to the analytical analysis found in [29]. The model used for quadruped kinematics in [29] consists of a pair of bipeds linked by a rigid body. Gaits are generated by varying the phases between the front and rear bipeds in a fashion similar to what here is *inferred* by *Gaitor*.

## 4.2 Latent Space Evolution During Terrain Traversal

The full perceptive pipeline of *Gaitor* is deployed on the real ANYmal C quadruped and its response to changes in terrain is evaluated. For these experiments, the operator selects the trot gait and the robot is commanded forward to climb a $12.5\,\mathrm{cm}$ platform. It is observed that the planner adjusts the robot's swing lengths and heights in response to the terrain. This is shown in Fig 5, where the planner's trajectories and the footswing characteristics are plotted during three distinct phases of terrain traversal. The first phase is normal locomotion on flat ground (phase I), the second is a short transient (phase II), where the front feet step up onto the palette and the final is a new steady state (phase III), where the robot has its front feet on the terrain and its rear feet on the ground. In

|  | Dynamic Gaits (Trot) | Trot (Flat) | Trot (Climb) | Crawl |
|---|---|---|---|---|
| RMSE [rad] | 0.021 | 0.012 | 0.013 | 0.058 |
| Swing Height [cm] | $8.5 \pm 0.40$ | $8.3 \pm 0.58$ | $9.68 \pm 4.15$ | $5.42 \pm 0.65$ |
| Swing Length [cm] | $14.6 \pm 3.99$ | $10.4 \pm 0.53$ | $13.9 \pm 1.65$ | - |

Table 1: The feasibility and gait characteristics for *Gaitor* and *Dynamic Gaits* used to generate the training data. *Gaitor's* gait characteristics and tracking root-mean-squared error (RMSE) values are recorded during four modes of operation: trot on flat ground, trot whilst climbing the terrain, crawl gait, and the dynamic crawl/pace gait. The RMSE values are measured between the joint-space trajectories of the *Gaitor* or *Dynamic Gaits* and the whole-body controller's (WBC) joint trajectories during a $5\,\text{s}$ interval. The WBC uses a centroidal dynamics model to optimise the torque values. The lower the RMSE, the closer the VAE trajectories are to optimality under the assumptions of the WBC. All measurements are generated from experimental logs from real-robot experiments. The swing lengths are not measured for crawl and dynamic crawl as the robot is operated on the spot during these runs. Both trot data come from the terrain climb experiment, see Sec. 4.2.

phase I in Fig. 5, the latent space trajectory is broadly symmetrical and *Gaitor* produces locomotion with footswing heights and lengths of $(8.30 \pm 0.58)\,\text{cm}$ and $(10.40 \pm 0.53)\,\text{cm}$ respectively. During the climb, the planner more than doubles the nominal horizontal displacement of the latent space trajectory from 1.98 units to 4.38 units, see the second row of Fig. 5. This results in a large increase in the swing length to $(13.90 \pm 1.65)\,\text{cm}$, but only a modest increase in swing height of $(9.68 \pm 4.15)\,\text{cm}$ in the robot's base frame. This increases the robustness of the foothold selection as this footfall location is further from the edge of the step. More details on elongating footswing length in response to terrain is provided in Sec. A.5.4. In phase III, the latent trajectory returns to a more symmetrical shape similar to phase I. This is expected once the front feet have climbed the terrain, the locomotion is similar to flat ground operation.

### 4.3 Feasibility And Comparison To Expert

To quantify the feasibility of the *Gaitor's* perceptive locomotion, we compute the joint-space error between the input and the output of the WBC. The root-mean-squared error (RMSE) is used to measure the optimality of the VAE's trajectories with respect to the WBC's centroidal dynamics [30]. The lower the RMSE value, the closer the VAE's trajectory is to satisfying the WBC's optimal solution. A $5\,\text{s}$ window during flat ground operation and the climb phase (middle panel in Fig. 5) is compared to the dataset RMSE in Table 1. The dataset uses the *Dynamic Gaits* [28] formulation, which is designed to operate with the WBC. The RMSE for the VAE during both flat ground and climb phases is lower than that of *Dynamic Gaits* (see Table 1) showing that *Gaitor's* trajectories are feasible for the WBC to track with little to no adjustments. In contrast, the WBC makes minor adjustments to the trajectories from *Dynamic Gaits*, which accounts for the larger RSME values.

## 5 Conclusions And Limitations

In this paper, we explored the opportunities afforded by learning a single, disentangled representation for locomotion across gaits to encode correlations between separate skills. In doing so, we discovered that the learnt latent-space embeds not only each individual gait into the space, but also orders each gait representation such that continuous transitions via novel intermediary contact schedules are discovered from traversing the learnt latent-space. *Gaitor* incorporates a simple terrain encoding which adapts the structure of the latent space in conjunction with a learnt planner that adjust the robot's gait characteristics. The latent-space structure changes to represent different gaits during transition and adapts the gait characteristics in response to the terrain encoding.

A limitation of *Gaitor* is the requirement of high-quality expert demonstrations. This is a limitation currently of all learning-from-demonstration methodologies and is lessened by the availability of quality locomotion controllers designed for specific tasks or gaits. In conclusion, we show that learning a single unified latent representation for locomotion across gaits results in a methodology capable of continuous gait transitions and perceptive locomotion to traverse significant obstacles.

## Acknowledgements

This work was supported by a UKRI/EPSRC Programme Grant [EP/V000748/1]. We would also like to thank SCAN for use of their GPU acceleration facilities. Finally, we thank Daniele De Martini, Jack Collins and Anson Lei for their useful discussions.

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

# A   Appendix

In this section, we include additional details and analysis of *Gaitor*. To begin with, a comparison between *Gaitor* and previous work *VAE-Loco*, which forms inspiration for this work is made. Following this, we discuss some details concerning deployment of the *Gaitor* as a planner on the real ANYmal C quadruped. Next, details concerning creating the dataset is provided followed by detail on the hyper-parameters used during the experiments. This appendix concludes with further interpretation of the results and comparison of *Gaitor's* capabilities in light of current state-of-the-art.

## A.1   Relationship of *Gaitor* to *VAE-Loco*

As mentioned in the main body of the text, the architecture of *Gaitor* is inspired by *VAE-Loco* [14] and the similarities and improvements of *Gaitor* over *VAE-Loco* are discussed here. *VAE-Loco* uses a VAE augmented with a performance predictor to estimate contact states to learn a latent space for *blind* locomotion on flat ground only. *Gaitor* integrates a perception pipeline for uneven terrain traversal and uses a learnt planner to generate adaptive trajectories in the latent space using a polar coordinate framework to adjust the robot's gait characteristics (footswing lengths and heights) to traverse terrain robustly. As a result, a phase-based relationship across all the gaits seen during training is learnt in *Gaitor*. This is used to find continuous transitions between distinct gait types via novel and inferred contact schedules. *Gaitor* includes a perception pipeline and a learnt planner in the latent space meaning that *Gaitor* can generate terrain-aware locomotion trajectories.

## A.2   Dataset Generation

Gaitor is trained using a narrow set of skill demonstrations. The data used to train all components are generated using *RLOC* [16]. This employs a vision-based RL footstep planner and uses *Dynamic Gaits* [28] to solve for the task-space trajectories. The inputs to the controller are the desired base heading $\mathbf{a}_k$ and the raw depth map provided by the robot. We generate roughly $30\,\mathrm{min}$ of data of the robot traversing pallets of varying heights up to $12.5\,\mathrm{cm}$ in both trot and crawl. The pace gait is unable to traverse uneven terrain so data for pace are gathered on flat ground. Our implementation of *Dynamic Gaits* is unable to pace stably at speeds greater than $0.1\,\mathrm{m/s}$ in simulation and is unstable at all speeds on the real robot. Therefore, we do not deploy any pace gaits.

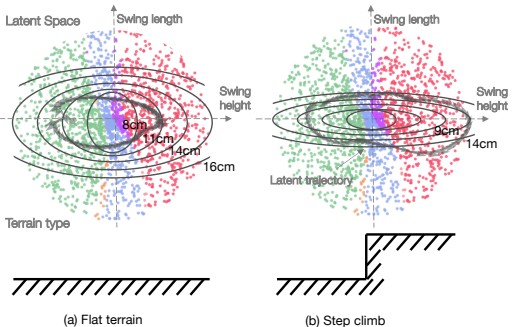

Figure 6: Level sets are plotted in latent space where isocontours decode to equal swing heights and lengths. The latent-space trajectories and isocontours are plotted during flat-ground operation (a) and as the robot approaches the step (b). The latent-space structure prioritises swing length during climb.

## A.3   Creating a Robot-Centric Depth Map

The process of creating the 2.5-dimensional height map which forms the input to the terrain processing pipeline is explained here. On the ANYmal robot, the height map is created using four depth cameras around the robot and the process to fuse the depth images into the height map is detailed in [23] and [24]. In general, the elevation mapping process requires any point cloud information from a camera source in a known frame and ideally the pose of the robot from a state estimator. The process then fuses the point cloud data and uses the variance of the pose estimates as a metric to choose which points in the point cloud are more reliable. The output of this process is the height map which we use to guide the learnt planner in latent space in order to react to uneven terrain.

In future work, alternative input media will be investigated to infer additional information about the robot's surroundings. For example, the cameras output RGBD images which could be used to infer the height around the robot directly as well as the contact properties of the terrain. These properties could inform the VAE about the friction of the terrain around the robot so that the robot can adjust its gait characteristics to maximise its stability.

## A.4  Architecture Hyper-parameters And Realtime Deployment

The VAE's encoder and decoder frequencies are set to $50\,\text{Hz}$ and $400\,\text{Hz}$. The input history contains $N = 80$ timesteps, whilst the output predicts a small $M = 20$ step trajectory. All networks are 256 units wide and three layers deep with the robot latent-space and the terrain latent-space both of dimensionality $\mathbf{z}_r(k) \in \mathbb{R}^{10}$ and $\mathbf{z}_g(k) \in \mathbb{R}^{10}$. The operator controlled base-heading action size is three units and the gait label is one unit. During terrain traversal, we utilise a desired swing-duration of $0.36\,\text{s}$.

*Gaitor* operates on the CPU of the ANYmal C in a real-time control loop. There is a strict time budget of $2.5\,\text{ms}$ for $400\,\text{Hz}$ control. Therefore, we deploy *Gaitor* using a bespoke C++ implementation and vectorised code.

## A.5  Further Analysis of Results

Analysis and interpretation of the results are included here. The latent space structures associated with the transitionary gaits are investigated. The information encoded into the other eight latent dimensions is also discussed. As mentioned in the results section (Sec. 4.2), *Gaitor* automatically elongates the footswing length as the robot approaches terrain. The latent space structure during the climb phase is inspected to understand how changes in the gait characteristics during changes in terrain are encoded. Following this, the ability for the WBC to track *Gaitor's* trajectories is visualised during the climb phase.

### A.5.1  Intermediary Latent Spaces

As mentioned in the Results Section 4.1, the latent space restructures to represent different gait types. Five latent space structures and the corresponding gait schedules are shown in Fig. 7. The trot, crawl and pace gaits are shown as before accompanied by two intermediary gaits between trot and crawl and crawl and pace. The base action is used to reduce the robot's velocity before transitioning into the crawl and then the crawl/pace hybrid gait. The intermediary latent spaces show the latent space restructures smoothly as the gait type is varied.

### A.5.2  Higher-Order Latent Dimensions

The latent dimensions with the two lowest predicted variances encode the robot's swing height and lengths. This section discusses the information encoded into the other eight robot latent dimensions. The means and variances of these latent dimensions are close to the prior distribution with a zero mean and close to unit variance. Decoding oscillatory signals in these latent dimensions reveal slight movements in the quadruped's joints. These joint motions are small and not easy to interpret. These motions are similar to the high-frequency components of a Fourier transform applied to the locomotion trajectories.

### A.5.3  Feasibility Metrics Of Gait Switching

The quality of the gaits produced by *Gaitor* is measured using the joint-space error between input and output of the whole-body controller (WBC) measured over a $5\,\text{s}$ window sampled at $400\,\text{Hz}$. The WBC is an optimisation-based tracking controller and is able to adjust the VAE's output to satisfy its dynamics constraints outlined in [31]. The root-mean-squared error (RMSE) is used as a measure of how optimal the VAE's trajectories are with respect to the dynamics of the WBC. The RMSE for the trot, crawl and dynamic pace/crawl gaits are reported in Table 1. In all cases the tracking error

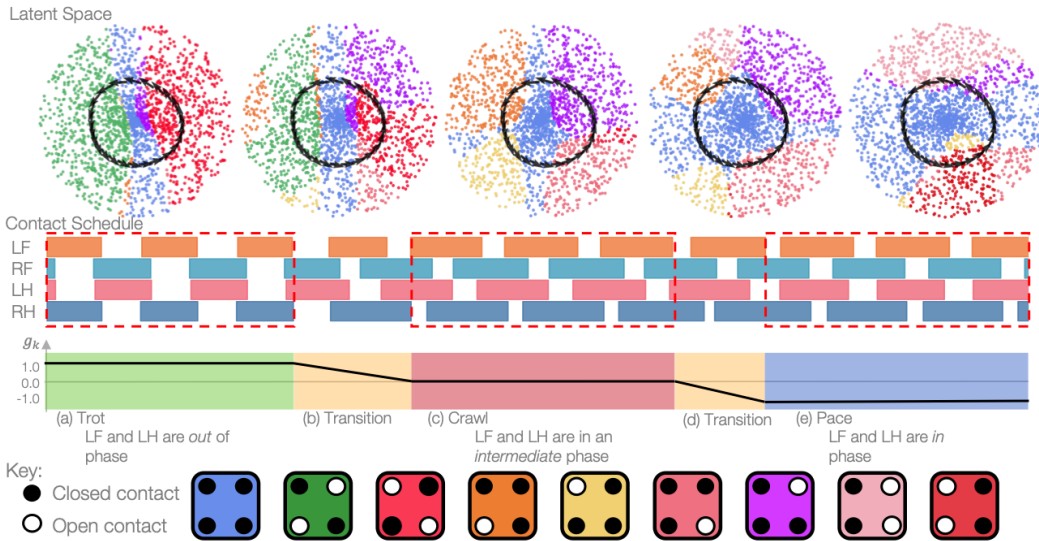

Figure 7: The latent representation of each distinct gait, trot, crawl, and pace and the intermediaries between the distinct gaits are plotted. Time increases from left to right. The top row shows five latent spaces starting with trot on the left and ending with pace on the right and crawl in the middle. The latent space images between trot, crawl and pace represent the transitionary gaits inferred by the model. The coloured dots represent the contact states of the robot and a key is provided at the bottom of the figure. We include the corresponding contact schedule with each latent space.

is low (less than $6 \deg$), and is larger for the crawl and dynamic crawl gaits. The larger joint RMSE results from a small discrepancy between the robot's actual and desired heights when operating in the crawl gaits.

### A.5.4 Latent Space Structure During Climb

During the climb (phase II), *Gaitor* increases the footswing length significantly more than the footstep height. This is investigated by inspecting how the encoding of the gait characteristics vary as the terrain changes. To visualise this, level sets in latent space are plotted in Fig. 6. Here, isocontours decode to locomotion trajectories with equal footswing height and length relative to the robot's base frame. For example, decoding the inner most contour in Fig. 6 (a) results in locomotion with a footswing height and length of $8 \mathrm{~cm}$ (the swing length is measured relative to the base frame). For both flat-ground operation and the climb phase, the latent-space trajectories follow the isocontours. Crucially, as the edge of the pallet is encoded, the latent-space structure adapts such that a small dis-

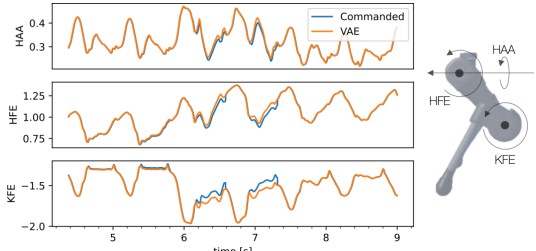

Figure 8: The joint-space trajectory in radians for the right front leg as it climbs terrain. The VAE's joint space trajectory is tracked well by the WBC's commanded values meaning that the VAE's output is optimal with respect to the WBC's dynamics formulation. A leg is plotted next to the tracking plots and annotated with joint names.

placement in the vertical axis of the latent space ($z_1$ dimension) decode to much longer footsteps. This is reflected in the planner's trajectory which adjusts similarly to the latent-space structure to follow the deformation of the isocontours. If the flat-ground latent-trajectory is used during terrain traversal, the robot does not lift its feet high enough over the step and the feet collide with the step. However, *Gaitor's* planner reacts to the oncoming terrain and produces a large footswing height with clearance of $4.6 \mathrm{~cm}$ above the edge of the pallet. In conclusion, the planner is necessary during

| Approach | RL vs Representation Learning | Discovers Intermediate Skills | Explicit vs Implicit Representation |
| --- | --- | --- | --- |
| Caluwaerts et al. [11] | RL (Bottom-Up) | No | Implicit |
| Hoeller et al. [6] | RL (Top-Down) | No | Implicit |
| Yang et al.[13] | RL + MPC | No | Explicit Inductive Bias |
| **Gaitor** | Rep. Learn | Yes | Explicit Inferred |

Table 2: The characteristics and differentiating features of *Gaitor* are compared to a range of approaches. Bottom-up RL methods such as [11] tend to employ black-box models to learn a single distribution over multiple locomotion skills. Top-down RL refers to hierarchical approaches [6] to locomotion, which use a high-level planner to select a particular skill from a series of sub-policies optimised for one particular gait. RL and MPC methods in combination [13] tend to use RL to solve for trajectories constrained by difficult to model parts of the robot's dynamics and use model-predictive control (MPC) to generate task-space trajectories. In contrast, *Gaitor* learns a unified representation across multiple gaits and in doing so is uniquely able to automatically discover new intermediate gaits and this ability is compared between approaches in the second column. The third column compares each method's representation for locomotion. The first two RL methods have an internal implicit representation, whilst the Yang et al. [13] and *Gaitor* use the gait phase as explicit representation. The explicit representation of gait phase naturally leads to sharing of gait-specific correlations across skills and facilitates gait transitions as described in Sec. 4.1. However, Yang et al., use the gait phase as an inductive bias and a decision variable during training and execution. In contrast, *Gaitor* infers the gait phase across multiple gaits from expert examples of the trajectories.

terrain traversal in order to fully exploit the latent-space structure, which usefully deforms as the robot approaches terrain.

### A.5.5 Tracking Results During Climb

In addition, the VAE's joint trajectory, which is the input to the WBC and the WBC's output (the commanded values) for the right-front leg are plotted in Fig. 8. The tracking error is insignificant (less than one degree) except during the initial touch down phase. This small increase in RMSE is attributed to vision error. The height of the step is slightly underestimated by the robot's perception module, and does not affect the stability of the locomotion. After a subsequent footswing, the VAE adjusts and the tracking error remains extremely low.

### A.6 Gaitor's Capabilities in Relation to Other Methods

The characteristics of Gaitor and key differentiators are compared to current methods in Table 2. As shown in Sec. 4.1, *Gaitor* is able to learn a unified representation across multiple gaits. In the same section, it is shown that this naturally yields continuous and smooth transitions between gaits in an interpretable way. Top-down hierarchical methods such as [6] use multiple independent policies and learn a planner to select the correct controller as required. This ignores correlations between the gait types and continous gait switching via intermediate gaits is impossible. Caluwaerts el al. [11] learn a multi-modal distribution policy over gait skills using a large transformer model. This does not produce an explicit representation, but an implicit one internal to the transformer.

In contrast, *Gaitor* creates a dense and explicit latent-representation and once trained this representation facilitates the automatic discovery of useful intermediary gaits leading to smooth and continuous gait transitions in an interpretable way as seen in Sec. 4.1 and Sec. 4.2. In particular, it is demonstrated in Sec. 4.1 that *Gaitor* infers the correlations, namely the phase relationships, between the gait types and uses these to structure the latent representation. In essence, *Gaitor* has inferred an inductive bias from expert data that is typically used in the formulations of other methods. For example, [12, 13] uses the gait phase as an explicit control parameter to generate blind gait transitions. In contrast, *Gaitor* infers the gait phase relationship across gaits and builds this directly into its locomotion representation.

