# OpenReview forum: "Gaitor: Learning a Unified Representation Across Gaits for Real-World Quadruped Locomotion"
_robot-learning.org/CoRL/2024/Conference — CoRL 2024_

### Official Review · Reviewer_CBEd · 2024-06-29
**Better illustration is needed.**

**Originality:** 3
**Technical Quality:** 3
**Clarity Of Presentation:** 3
**Potential Impact:** 4
**Recommendation:** 3
**Confidence:** 3

**Review:**

Strengths:

- The topic is interesting. Constructing a good latent representation is crucial for regenerating motions in the dataset.
- The visualizations, especially the 2D trajectory plots, are well-presented.
- The latent space analysis is adequately conducted.


Weaknesses:

- Clarity: The illustrations need improvement. In Figure 1, the use of many colors makes it difficult to find the correspondence between the latent space and gait patterns. Figure 2 is also hard to understand from the caption; the symbols seem to appear without explanation. In Figure 2(A), the robot state encoder outputs a 3D vector, but the third dimension is not clearly explained, nor is its difference from the previous two vectors. Additionally, in Line 113, the user-controlled vector is not well-explained.

- Lack of Motivation Illustration: The motivation of the paper is not clearly explained. The importance of having a good representation and how different representations can affect the robot's movement need to be better articulated.

- Missing Related Works: Two important related works addressing the problem of constructing locomotion latent representations are missing [1][2]. It would be beneficial to compare your method to these existing methods.

[1] Starke, et al. DeepPhase: Periodic Autoencoders for Learning Motion Phase Manifolds.

[2] Li, et al. FLD: Fourier Latent Dynamics for Structured Motion Representation and Learning.

**Quality Of The Limitations Section:**

2

**Questions For Rebuttal:**

In the rebuttal, I would like to see a more detailed explanation of the motivation for constructing a good representation and how it affects the robot's performance. Additionally, I would appreciate a clearer explanation of the figures.

**Robotics Focus:**

4

**Summary Of Paper:**

This work proposes a new embedding approach that projects different quadrupedal robot locomotion gaits into a disentangled, 2D latent space. The approach consists of an encoder-decoder structure for reconstructing the robot's future joint and contact states. It also involves a terrain VAE that learns to use terrain information to adjust the robot's future desired pose. The learned representation can then be reused for planning by training a planner to output a vector that adjusts the latent variable. Simulation experiments have been conducted to analyze the latent space structure and how it evolves according to different terrains. Hardware experiments have also been conducted to validate the approach.

**Summary Of Recommendation:**

The author provide extra illustration during rebuttal stage that makes the paper's method contribution easier to be captured.

---

### Official Review · Reviewer_jDLg · 2024-07-20
**Learning a disentangled 2D representation of quadruped gaits with a VAE framework**

**Originality:** 4
**Technical Quality:** 4
**Clarity Of Presentation:** 4
**Potential Impact:** 3
**Recommendation:** 4
**Confidence:** 3

**Review:**

An interesting and well written paper. The authors present a VAE model augmented with perception-based terrain information to learn a unified and interpretable 2D latent space describing footswing height and length. This model admits continuous transitions between types of gaits, where intermediate gaits are inferred/discovered by traversing the 2D latent space. The paper is written clearly and the originality of the work is fair.

The strength of this paper is its significance to the robotics learning community. Access to this interpretable, disentangled, 2D latent representation can strengthen the communities understanding regarding gait transitions. In particular, the corroboration of analytic results in citation [26] is a brilliant example of this. This result shows the power and place that interpretable learning frameworks have in robotics/control. It strengthens both the analytic and the learning-based communities' understanding when corroborations such as these are found in research. This corroboration bolsters the impact that interpretations and understandings of gait transitions formed through analysis of Gaitor's latent space can have on the community. Since Gaitor is able to infer/generate gaits, inferred locomotion patterns can be studied to improve or inform future analytic modelling.

**Quality Of The Limitations Section:**

3

**Questions For Rebuttal:**

Questions:
- Do the authors have any insights into what the remaining 8 dimensions of the latent space represent in terms of locomotion? Are there variables describing left/right movement of the leg during the footswing, rather than purely up/down? Left/right or lateral movement of limbs can become important for traversing difficult or uneven terrain, so would strengthen the framework if it were included.
- Out of curiosity, do the authors expect that the terrain encoding can extend not just to height, but to other terrain properties such as slip/friction coefficients, etc, so that the community may understand gait transitions between such terrains in a similar way? Would the latent space representation then require higher dimension, etc?

Small comments:
- in the caption of Figure 3 \mathbf{x}_g is written rather than \mathbf{X}_g
- In figure 4, main text refers to colours blue and teal to indicated right front and left hind contact state vs all four feet in contact, but it is hard to tell apart blue from teal in the figure (if in fact there is any teal at all), and there are some pink points which are unidentified in text.

**Robotics Focus:**

4

**Summary Of Paper:**

By learning a disentangled 2D representation of gaits, where each variable represents the gaits horizontal and vertical length, continuous gait transitions can be planned, allowing for smooth traversal of terrain with varying height. Furthermore, since the 2D latent space is interpretable, insights can be made regarding the continuous gait transitions. These insights corroborate analytical results/analysis regarding gait transitions.

**Summary Of Recommendation:**

This paper presents a novel learning framework for quadruped locomotion that is interpretable as footswing height and length. The framework is evaluated on hardware (ANYmal) and demonstrates strong significance to the community regarding understanding gait transitions in quadruped robots. Evaluation is based around this contribution rather than demonstrating state-of-the-art locomotion, and is well presented.

---

### Official Review · Reviewer_sMp4 · 2024-07-21

**Originality:** 3
**Technical Quality:** 4
**Clarity Of Presentation:** 5
**Potential Impact:** 3
**Recommendation:** 3
**Confidence:** 3

**Review:**

The paper is well-written and clearly organized, making it easy to follow. While there have been several related prior works that have been positioned in the domain of directly controlling end-effector trajectories and gait schedules, the current method affords certain additional benefits.

Strengths:
- Interpretability of the latent embeddings using the first two dimensions of the latent space. Additionally, given that these two dimensions are disentangled/independent allows for more fine-grained control of the robot’s gait.
- Emergence of smooth and continuous gait transitions, even though these were not observed during training.

Weaknesses:
- The terrain encoder relies on the perception module consisting of multiple cameras to create the height map - while this is applicable in the current setting, it may be difficult to obtain the required information in the absence of a similar sensor configuration.
- The method has currently been tested only on trot, pace, and crawl, and transitions between them. It would be interesting to see the method being extended to other gaits, including higher velocity gaits.

**Quality Of The Limitations Section:**

3

**Questions For Rebuttal:**

- Can this approach be extended to more dynamic motions like box jumps or jumping across gaps?
- It would be nice to see a comparison of motion quality, especially transitions, to other methods cited by the authors such as Wu et al (Learning Multiple Gaits within Latent Space for Quadruped Robots) or hierarchical methods such as the one proposed by Yang et al (Fast and efficient locomotion via learned gait transitions).
- Certain gaits can only be realized in certain speed ranges - how is this being enforced?
- Is the planner capable of recovering from external perturbances?

**Robotics Focus:**

4

**Summary Of Paper:**

The proposed method learns an interpretable, disentangled, 2D representation of quadrupedal locomotion gaits by leveraging correlations between gait types that allows achieving smooth and sometimes novel transitions during deployment. By combining information from the perception module and utilizing a terrain encoder, the authors also learn a planner that takes as input desired gait parameters, base velocity commands, and the terrain encoding, and plans in the learned latent space to adapt to changing terrain in a closed-loop manner.

**Summary Of Recommendation:**

The paper proposes a novel and interpretable method for achieving different gaits on a quadrupedal robot and smoothly transitioning between them. The proposed method and the authors' claims are backed up by real world results.

---

### Author Rebuttal · Authors · 2024-08-12

We take this opportunity to thank all the reviewers for their time and for their suggestions. We have implemented them in the revised manuscript and these improvements have raised the quality of our submission.

The changes made to the manuscript are in blue and these are summarised here. We have adjusted Fig. 1 so that it better reflects the contribution of the paper. The figure reflects that we learn a single latent representation for locomotion across gaits resulting in perceptive locomotion and continuous gait transitions. We have also simplified Fig. 2 and made the flow between the elements easier to understand. We have added two papers to the related work sections which use specific Fourier-based architectures to learn rotational locomotion dynamics in a latent space. We contrast these methods to VAE-Loco, which forms the inspiration for our submission Gaitor. VAE-Loco uses the contact dynamics to learn rotational trajectories in latent space for locomotion. We have amended the colour schemes of the latent space figures in the paper so that there is better contrast between contact clusters. We swap the teal colour with a deeper green hue. We explain how the robot-centric depth map is created and point the reader to two papers where this process is introduced. We also explain exactly which elements of the architecture are user-controlled inputs and what is inferred. The transitionary latent space structures have been moved from Fig. 1 to a new figure in the Appendix Fig. 7. This new figure has all five latent space structures referring to trot, trot/crawl, crawl, crawl/pace and pace. We clarify that the robot’s legs swing in the direction of the desired base-heading meaning that the robot naturally follows velocity commands provided by the operator. We also discuss what is encoded in the other eight latent dimensions in the appendix. We shall also add in a short description of how to adjust the robot's base speed in order to transition between gaits when there is a large difference between gait operating speeds.

We thank the reviewers again for their time and look forward to hearing from them soon.

---

### Decision · Program_Chairs · 2024-09-04

**Decision:**

Accept

**Comment:**

All reviewers agree on the significance of the approach and the interesting results allowing the emergence of smooth gait transitions without simulation training. This proves the effectiveness of the proposed disentangled representation for the locomotion task.

The weaknesses seem to be easily solvable: on one hand, it would require a better description of the limitations, specifically on the required setup and on how this could be extended to not just height but also to other terrain properties. On the other hand, the paper would need a bit of improvement on the presentation side, with better visualization of the key ideas (indeed the images are a bit complicated and don't convey directly the major sights of the approach). Finally, the authors should respond in depth to the questions of the reviewers and add the missing citations (or properly argue why they are not relevant).

===

All reviewers are satisfied with the author's response during the rebuttal phase, therefore I'll mark this paper for acceptance.